# Preterm Infants on Early Solid Foods and Iron Status in the First Year of Life—A Secondary Outcome Analysis of a Randomized Controlled Trial

**DOI:** 10.3390/nu14132732

**Published:** 2022-06-30

**Authors:** Margarita Thanhaeuser, Fabian Eibensteiner, Margit Kornsteiner-Krenn, Melanie Gsoellpointner, Sophia Brandstetter, Renate Fuiko, Ursula Koeller, Wolfgang Huf, Mercedes Huber-Dangl, Christoph Binder, Alexandra Thajer, Bernd Jilma, Angelika Berger, Nadja Haiden

**Affiliations:** 1Department of Pediatrics and Adolescent Medicine, Comprehensive Center for Pediatrics, Medical University of Vienna, 1090 Vienna, Austria; margarita.thanhaeuser@meduniwien.ac.at (M.T.); fabian.eibensteiner@meduniwien.ac.at (F.E.); margit.kornsteiner.krenn@gmail.com (M.K.-K.); sophia.brandstetter@meduniwien.ac.at (S.B.); renate.fuiko@meduniwien.ac.at (R.F.); mercedes.huber-dangl@meduniwien.ac.at (M.H.-D.); christoph.a.binder@meduniwien.ac.at (C.B.); alexandra.thajer@meduniwien.ac.at (A.T.); angelika.berger@meduniwien.ac.at (A.B.); 2Department of Clinical Pharmacology, Medical University of Vienna, 1090 Vienna, Austria; melanie.gsoellpointner@meduniwien.ac.at (M.G.); bernd.jilma@meduniwien.ac.at (B.J.); 3Department of Laboratory Medicine, Klinik Hietzing, Wiener Gesundheitsverbund, 1130 Vienna, Austria; office.koeller@gmail.com (U.K.); wolfgang.huf@gesundheitsverbund.at (W.H.)

**Keywords:** preterm infant, solid foods, iron status, iron intake

## Abstract

Introduction of solid foods and iron status in the first year of life of preterm infants are highly discussed topics. The aim of this study was to examine whether two timepoints of introduction of standardized solid foods in preterm infants have an impact on ferritin and other hematologic parameters important for iron status in the first year of life. This is a secondary outcome analysis of a prospective, randomized intervention trial in very low birth weight (VLBW) infants randomized to an early (10–12th week corrected age) or a late (16–18th week corrected age) complementary feeding group. Iron status was assessed with blood samples taken at 6 weeks, 6 months, and 12 months corrected age. In total, 177 infants were randomized (early group: *n* = 89, late group: *n* = 88). Ferritin showed no differences between study groups throughout the first year of life, as did all other parameters associated with iron status. At 12 months corrected age, the incidence of iron deficiency was significantly higher in the early feeding group. There is room for improvement of iron status in VLBW preterm infants, regular blood checks should be introduced, and current recommendations may need to be a reconsidered.

## 1. Introduction

In the third trimester of pregnancy, most fetal iron is transferred from the mother to the fetus leading to low iron stores at birth in preterm infants as pregnancy is interrupted [1,2]. After birth, rapid growth and phlebotomy losses lead to a high risk of iron deficiency (ID) or iron deficiency anemia (IDA) [1]. The risk is even higher in infants born small for gestational age (SGA) or those with intrauterine growth retardation (IUGR) [3]. Other factors which negatively influence the iron status of infants are low maternal iron status, gestational diabetes, smoking habits of the mother, male sex, and breastfeeding [3].

Iron deficiency leads to poor short- and long-term health and neurodevelopmental outcomes in preterm infants [4]. Infants with iron deficiency are at risk of altered brain structure and function with impaired cognitive, motor, and neurobehavioral development due to altered metabolism and neurotransmission as well as effects of iron deficiency on myelination and synaptogenesis during brain growth [5,6]. Therefore, supplementation of iron is needed in the early life of preterm infants [4,7]. Recommendations for preterm infants during hospital stay vary from 1 to 9 mg/kg/d depending on supplemental erythropoietin therapy [8,9,10]. For the post discharge period, the European Society for Gastroenterology, Hepatology, and Nutrition (ESPGHAN) recommends to supplement 2–3 mg/kg/d in very low birth weight (VLBW) infants until 6–12 months with a close monitoring of serum ferritin levels to avoid iron deficiency but also overload [7,11].

During the first year of life, iron intake and status of infants changes as solid foods are introduced. As soon as iron-rich solid foods are fed on a regular basis, it is likely that iron supplementation in preterm infants may be stopped due to the higher intake. Jonsdottir et al. showed that early complementary feeding has a small positive effect on iron status of term infants with higher serum ferritin levels in the early feeding group at 6 months [12]. Hemoglobin and MCV levels did not differ between groups [12].

How early complementary feeding affects the iron status of preterm infants is an unexplored field so far. The aim of this study was to examine whether two different timepoints of introduction of solid foods in preterm infants have an impact on their iron status in the first year of life with a primary focus on serum ferritin levels and the following secondary outcomes: hemoglobin, hematocrit, red blood cell count, red blood cell indices (MCV, MCH, MCHC), transferrin, iron, transferrin saturation, and soluble transferrin receptor. Due to its association with altered neurodevelopmental outcome, incidence of iron deficiency, anemia, and iron deficiency anemia was collected as well.

## 2. Materials and Methods

This study was a pre-specified secondary outcome analysis of a prospective, randomized, two arm intervention trial in preterm infants on early solid foods. The study was conducted from October 2013 to February 2020 at a level IV neonatal care unit of the Department of Pediatrics, Division of Neonatology, Pediatric Intensive Care and Neuropediatrics (Medical University of Vienna, Vienna, Austria). The study design and the primary outcome were recently published [13]. Inclusion criteria were a gestational age <32 weeks of gestation and a birth weight <1500 g. Infants with diseases affecting stable growth, i.e., necrotizing enterocolitis (NEC) with short bowel syndrome [14], Hirschsprung disease [15], chronic inflammatory bowel disease [16], bronchopulmonary dysplasia (BPD) [17], congenital heart disease [18], or major congenital birth defects or chromosomal aberrations were not eligible. Parents of infants were contacted at the outpatient clinic at their expected date of birth and randomized to an early complementary feeding group with an introduction of solid foods between the 10th and 12th week of life corrected for term compared to a late complementary feeding group with an introduction of complementary food between the 16th and 18th week of life corrected after informed consent was obtained from the parents. Permuted blocks were used for randomization (ratio 1:1, block size of six) using an online randomizer (www.randomizer.at, accessed on 22 June 2022) and infants were stratified according to feeding type (breast vs. formula/mixed feeding). Multiplets were assigned to the same group. At a recruitment status of 75%, an interim analysis detected a baseline imbalance (birthweight and gestational age) and randomization was switched to a baseline adaptive randomization with an additional stratification according to birthweight to correct the imbalance.

Sample size calculation was made using a two-sample t-test assuming a mean difference in body length at 12 months corrected age (primary outcome) of five percent (coefficient of variation of 11%, significance level of 0.05, 80% power) resulting in 76 patients per group.

Infants were fed age-appropriate standardized complementary food in addition to breastfeeding or formula until one year corrected for prematurity. Details on the standardized feeding boxes were previously described [13]. Infants with a weight <10th percentile at discharge received fortified breast milk (Aptamil FMS, Milupa Nutricia GmbH, Frankfurt, Germany) or preterm discharge formula until 52 weeks corrected age. Supplementation with vitamin D3 (650 I.E./day) was given until one year corrected age, and supplementation with iron (2–3 mg/kg/day, Ferrum Hausmann, iron oxide polymaltose complex, Vifor France, Paris, France) until iron-rich solid foods were fed on a regular basis [10]. Furthermore, infants received a multivitamin preparation (vitamins A, E, D3, B1, B2, B6, C, niacin, pantothenic acid; Multibionta, Merck Selbstmedikation GmbH, Darmstadt, Germany) until one year corrected age.

The primary focus of this secondary outcome analysis was serum ferritin levels throughout the first year of life. Secondary outcomes included further hematologic parameters important for iron status, i.e., hemoglobin (Hb), hematocrit (Hct), red blood cell count, red blood cell indices (MCV, MCH, MCHC), transferrin (TRF), iron, transferrin saturation (TFS), and soluble transferrin receptor (sTRF) assessed by serial blood samples taken at 6 weeks corrected age before the introduction of solid foods and at 6 and 12 months corrected age. Furthermore, incidence of iron deficiency (defined as ferritin < 40 mcg/L at 6 weeks corrected age and <12 mcg/L at 6 and 12 months corrected age), anemia (defined as Hb < 9 g/dL at 6 weeks corrected age and <10.5 g/dL at 6 and 12 months corrected age), and iron deficiency anemia (defined as ferritin < 40 mcg/L and Hb < 9 g/dL at 6 weeks corrected age and ferritin < 12 mcg/L and Hb < 10.5 g/dL at 6 and 12 months corrected age) were collected [11].

Ferritin was measured with Chemiluminescence Microparticle Immunoassay (CMIA, Abbott Alinity I, Abbott Park, IL, USA), Hb, Hct, red blood cell count, and red blood cell indices were all measured with the XE-2100 hematology system (Sysmex, Kobe, Japan), TRF with Immunoturbidimetry (Abbott Alinity c, Abbott Park, IL, USA), iron with Colorimetry (Abbott Alinity c, Abbott Park, IL, USA), and sTRF with Immunonephelometry (Siemens Atellica NEPH 630, Siemens, Berlin/Munich, Germany). Transferrin saturation was calculated from iron and transferrin (quotient of iron and transferrin multiplied by 0.709).

Baseline characteristics and parameters on neonatal morbidity are shown in Table 1. Small for gestational age was defined as birth weight <10th percentile. During hospital stay, anemia was diagnosed as soon as transfusions or a therapy with erythropoietin were needed according to the recommendations of Shannon et al. [19]. Erythropoietin therapy (either 300 I.E./day i.v. or 700 I.E. s.c. three times a week) was given in stable patients (“feeders and growers”) along with folic acid (0.1 mg/kg) and a higher intake of iron (9 mg/kg/d) [9,20].

### 2.1. Study Visits

Participants were invited to study visits at expected due date, 6 weeks, 12 weeks, 6 months, and 12 months corrected age along with clinical routine care visits at the neonatal outpatient clinic. Anthropometric measurements were collected at every visit, and results were published previously [13]. Blood samples were taken at 6 weeks corrected age before solid foods were started, as well as at 6 and 12 months corrected age, respectively.

Written informed consent from one parent was sufficient due to low risk for participants. The study was approved by the institution’s ethics committee (EK: 1744/2012) and registered at clinicaltrials.gov (NCT01809548).

### 2.2. Statistical Analysis

In general, for ordinal and nominal data, absolute and relative frequencies were calculated. Mean and standard deviation, or median and interquartile range (IQR), were calculated for continuous variables. Iron supplementation was expressed as mg/kg/day being calculated as cumulative iron supplementation from discharge until the respective visit (6 weeks corrected age, 6 months corrected age, 12 months corrected age) divided by body weight (in kg) at the respective visit and number of days of supplementation until the respective visit. To evaluate differences between groups in iron supplementation, a linear mixed effects model was fit through study group, gestational age at birth, sex, and nutrition at discharge (breastfed vs. formula).

For the evaluation of group differences of the primary outcome ferritin and the secondary outcome parameters transferrin, sTRF, iron, and transferrin saturation, a linear mixed effects model containing study group, gestational age at birth, sex, nutrition at discharge, cumulative iron supplementation adjusted for body weight (mg/kg), and erythropoietin supplementation was calculated.

For the evaluation of group differences in the secondary outcome parameters representative for red blood cell production (Hb, Hct, RBC, MCV, MCH, MCHC, reticulocytes relative), the model was extended by folic acid supplementation. To account for possible correlation between siblings of multiple births, an appropriate random intercept was introduced to all models. Results are reported as estimated marginal means, respective 95% confidence intervals (95% CI), and uncorrected *p*-values. For sensitivity analysis, a Mann–Whitney U test was utilized to calculate group differences on single occasions of very low patient numbers per group. Study group differences over clinically meaningful and established cut-offs for iron deficiency were evaluated by utilizing a mixed effects logistic regression through study group, gestational age at birth, sex, nutrition at discharge (breastfed vs. formula), erythropoietin supplementation, and cumulative iron supplementation adjusted for body weight (mg/kg). Again, a random intercept to account for possible correlation between siblings of multiple births was introduced to all models. For sensitivity analysis, a Fisher’s exact test was utilized to calculate group differences on single occasions of very low patient numbers per group. Differences in neurodevelopmental outcome were calculated with a linear mixed model (fixed effects iron deficiency, sex, gestational age at birth; random intercept multiple birth). Statistical analysis was conducted using R software (R Core Team 2020, www.R-project.org, accessed on 27 June 2022).

## 3. Results

### 3.1. Screening and Participants

In total, 177 infants were randomized between October 2012 and April 2020, with 89 infants to the early group and 88 infants to the late group.

### 3.2. Baseline Characteristics and Neonatal Morbidity

Detailed demographic parameters as well as information on neonatal morbidity and nutritional parameters were published previously [13], and parameters related to the iron status of infants are shown in Table 1. There were no significant differences between study groups.

### 3.3. Primary Outcome

Data of the primary outcome serum ferritin levels are shown in Figure 1. Serum ferritin levels showed no differences between study groups at 6 weeks, 6 months, and 12 months corrected age, respectively.

### 3.4. Secondary Outcomes

Results from the mixed effects model analysis on iron status are shown in Table 2. Supplemental iron intake decreased in the first year of life but stayed within the recommended range of 2–3 mg/kg/d for VLBW infants (Figure 2).

At 6 weeks corrected age, hemoglobin levels were significantly lower in the early group despite a significantly higher iron intake by supplements. Upon correction for multiple testing, none of these two differences remained significant. All other hematologic parameters where comparable between the two groups at 6 weeks, 6 months, and 12 months corrected age, respectively. At 12 months corrected age, only a small number of patients (early group *n* = 7, late group *n* = 16) still received iron supplementation indicating that differences between groups calculated by mixed model analysis should be interpreted carefully.

Incidences of iron deficiency, anemia, and iron deficiency anemia are shown in Table 2 as well. Due to low absolute numbers of events, differences were only calculated for iron deficiency, but not for anemia and iron deficiency anemia.

A mixed effects logistic regression did not show differences at 6 weeks and 6 months corrected age, respectively. The incidence of iron deficiency at 12 months corrected age was significantly higher in the early feeding group (*p* = 0.04, Fisher’s exact test *p* = 0.04), but not after correction for multiple testing. Due to low numbers of events, interpretation should be undertaken carefully.

Post-hoc, differences in iron status between infants according to feeding type and in those small for gestational age were calculated as well (Appendix A). Breast fed infants showed lower ferritin levels at 6 weeks corrected age. Furthermore, differences in neurodevelopment (Bayley-III) at 12 months corrected age between infants with and without iron deficiency at 6 weeks corrected age are shown in Appendix A. Infants with iron deficiency had higher motor and language composite scores.

## 4. Discussion

This secondary outcome analysis of a randomized controlled trial showed that the timepoint of introduction of solid foods had no impact on iron status in the first year of life of VLBW preterm infants; especially, serum ferritin levels were comparable between groups. A higher incidence of iron deficiency at 12 months corrected age in the early feeding group was revealed, but low numbers of events call for a cautious interpretation of this result. Age at weaning also did not show an influence on the primary outcome of the randomized controlled study, as growth was comparable between study groups. Regarding growth and iron status, early introduction of solid foods was safe in our cohort of infants with a mean birth weight <1000 g.

To our knowledge, data of VLBW preterm infants on early solid foods and their iron status in the first year of life have not been published so far. Randomized controlled trials in term infants raised hope for higher ferritin levels in infants with an early introduction of solids, as Jonsdottir et al. and Dewey et al. showed significantly higher ferritin levels in term infants starting solid foods at 4 compared to 6 months of age in a high- and low-income country, respectively [12,21]. Gupta et al. reported on complementary feeding of preterm infants at 4 versus 6 months of corrected age, but in preterm infants born at less than 34 weeks of gestation with a mean birth weight of 1600 g in a low-income country [22]. Serum ferritin levels and incidence of iron deficiency at 12 months of age was comparable between study groups with a trend to lower ferritin values and a higher incidence of iron deficiency in the early group, but this finding was not significant [22]. In our study, we could not detect significant differences in serum ferritin levels between study groups in infants with a mean birth weight <1000 g. Cooke et al. recently published that serial measurements of transferrin saturation and MCV are more sensitive measures of iron status in preterm infants than serum ferritin and hemoglobin [23]. However, no significant differences in MCV and transferrin saturation could be detected between study groups. Furthermore, infants with iron deficiency at 6 weeks, 6 months, and 12 months corrected age had comparable MCV and transferrin saturation values. Other hematologic parameters important for iron status were comparable between groups. This indicates that early introduction of solid foods in the vulnerable patient group of extremely low birth weight infants is safe concerning their iron status [13].

Incidence of iron deficiency defined as ferritin levels < 12 mcg/L was significantly higher in the early feeding group at 12 months corrected age, but due to the low numbers of events (early group: *n* = 8/62–13%, late group: *n* = 1/61–1%), interpretation of these results should be undertaken very carefully. At 6 weeks corrected age, the incidence of iron deficiency (ferritin levels <40 mcg/L) was considerably higher than later in the first year of life with 56% in the early and 63% in the late group, despite mean ferritin levels >40 mcg/L in both groups at the same age. The threshold of serum ferritin for the definition of iron deficiency changes within the first year of life with <40 mcg/L at 2 months of age and <10–12 mcg/L starting from 6 months of age [11]. This might explain the decrease in iron deficiency incidence between 6 weeks and 6 months corrected age. Nevertheless, around 90% of infants in both groups had anemia during their hospital stay and needed red packed blood cells or erythropoietin therapy but still especially those without therapy or on erythropoietin therapy only had depleted iron stores after discharge. This calls for further studies and a possible change in recommendations of iron supplementation for discharged VLBW preterm infants to optimize their iron status around term age and beyond. Regular checkups on blood count and ferritin levels early after discharge should be introduced to avoid iron deficiency as suggested by Domellöf et al. [7]. Moreover, particular attention should be paid to infants at higher risk of developing anemia and iron deficiency.

The current literature showed a higher incidence of anemia, iron deficiency, and iron deficiency anemia in breastfed infants [24]. Indeed, in our study, hemoglobin, hematocrit, and MCV levels were significantly lower and the incidence of anemia significantly higher in breastfed infants at 6 weeks corrected age as compared to formula fed ones, with no differences in ferritin levels, incidence of iron deficiency, or iron deficiency anemia (Appendix A). This indicates that at this timepoint anemia was more related to nutritional deficits such as under-provision of protein than to iron deficiency. It is important to mention that infants were stratified to breastfeeding or formula-feeding at discharge. At 6 and 12 months corrected age, no differences between formerly breastfed and formula-fed infants could be detected. Furthermore, infants born small for gestational age are at increased risk of developing iron deficiency. McCarthy et al. showed a significantly higher incidence of cord blood ferritin <76 mcg/L and <35 mcg/L in SGA infants [3]. Iron status of SGA infants was comparable to those born appropriate for gestational age in our study population during the first year of life, except for significantly lower hemoglobin levels at 6 months corrected age (Appendix A). Ferritin levels even were higher in SGA infants throughout their first year of life, but this finding was statistically not significant. Again, close monitoring of red blood count and ferritin levels during hospital stay and after discharge is important in all preterm infants but especially in those at risk for iron deficiency such as breastfed infants and those born small for gestational age. The current recommendations for iron supplementation for preterm infants by the ESPGHAN suggest a daily intake of 2 mg/kg/d in infants with a birth weight of 1500–2500 g from 2–4 weeks of life until 6 months of age or possibly longer depending on infant diet [11]. For VLBW infants, an intake of 2–3 mg/kg/d starting at two weeks of age until 6–12 months of age depending on infant diet with repeated measurements of serum ferritin levels during hospital stay and adaption of supplementation is suggested [11].

In our study, supplemental iron intake decreased in the first year of life but stayed within the European recommendation of 2–3 mg/kg/d for VLBW infants [11]. Still, iron stores early after discharge were depleted in both groups, although mean iron supplementation at 6 weeks corrected age was >3 mg/kg/d in both groups. Furthermore, during hospital stay, the iron supplementation was within the suggested range of 2–3 mg/kg/d, and more than 85% of infants in both groups received PRBC and/or erythropoietin therapy. This calls for an improvement of iron status in this vulnerable patient group including an adaption of current recommendations. Parents’ awareness regarding the importance of the iron status of their infants should be raised with the help of nutritional counseling after discharge before starting solid foods. Early introduction of meat products and iron-fortified complementary foods in combination with iron supplementation is important to prevent ID and IDA [25]. This needs to be addressed during parent counseling.

There is consensus in the literature that iron deficiency anemia is related to later poor neurodevelopmental outcome [4]. Iron deficiency anemia affects myelination, synaptogenesis, alters metabolism, and neurotransmission causing altered brain structure and function leading to impaired cognitive, motor, and neurobehavioral development [5,6]. In the present study, >50% of infants in both groups suffered from iron deficiency after discharge. However, infants who were iron deficient at 6 weeks corrected age showed significantly better scores in motor and language outcome at 12 months corrected age (Bayley-III test), which was unexpected (Appendix A). Cognitive scores were also higher in the group of iron deficient infants, but this finding was statistically not significant. Most studies reported on adverse outcomes especially in infants with IDA, which was the case in two infants of our cohort only [4,5]. Moreover, adverse outcomes of infants with IDA were measured at pre-school, school, or adolescent age and are not comparable with our cohort yet. As effects of iron deficiency without anemia on neurodevelopmental outcome are not sufficiently studied, follow up of our cohort is of high interest.

As the iron status is a secondary outcome it comes with various limitations. Power calculations were made for the primary outcome length; thus, the number of patients was possibly too low to detect a significant difference between study groups when it comes to the infants’ iron status. Many parents did not note and could not remember the exact date of end of iron supplementation.

## 5. Conclusions

The timepoint of introduction of solid foods had no impact on ferritin levels and other hematologic parameters important for iron status in the first year of life of VLBW preterm infants but showed a higher incidence of iron deficiency at 12 months corrected age in the early feeding group. There is room for improvement of iron status in VLBW preterm infants and current recommendations may need to be a reconsidered. Blood counts and serum ferritin levels should be measured on a regular basis before discharge and beyond with a special attention to breastfed infants and those born small for gestational age. Parents’ awareness regarding the importance of iron supplementation and early introduction of meat products and iron-fortified solid food should be raised by nutritional counseling.

## Figures and Tables

**Figure 1 nutrients-14-02732-f001:**
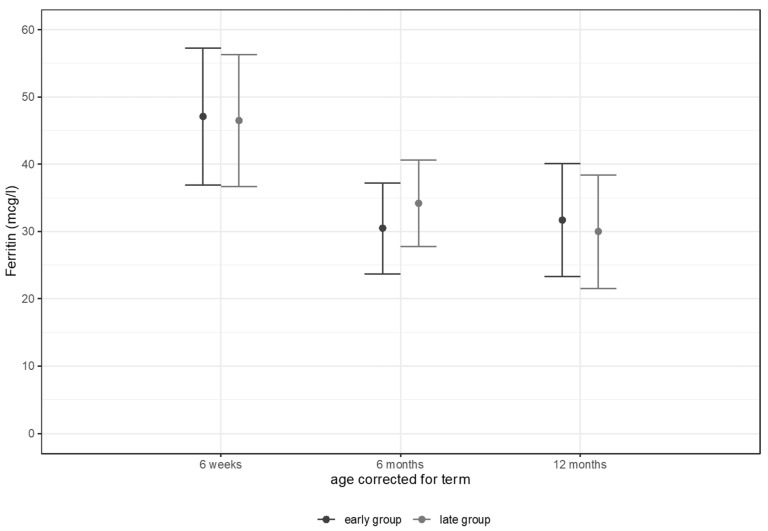
Ferritin levels (mcg/L) at 6 weeks, 6 months, and 12 months corrected age.

**Figure 2 nutrients-14-02732-f002:**
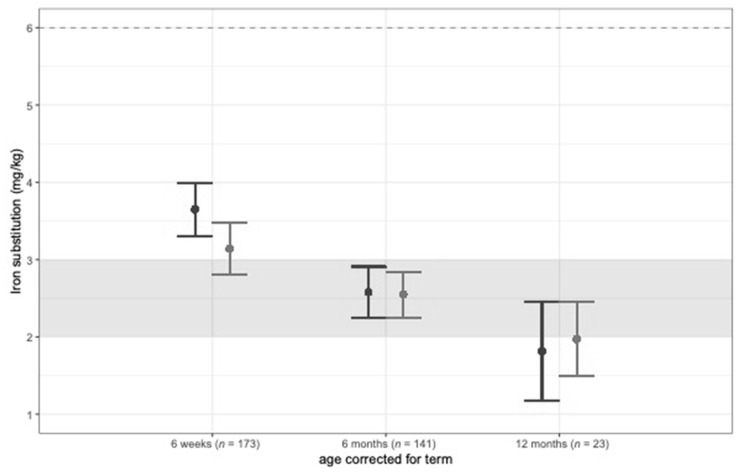
Iron supplementation (mg/kg/d) at 6 weeks, 6 months, and 12 months corrected age.

**Table 1 nutrients-14-02732-t001:** Baseline characteristics and neonatal morbidity.

Parameter	Early Group (*n* = 89)	Late Group (*n* = 88)
*Obstetric and parental parameters*		
Multiple pregnancy	32 (36)	28 (31.8)
Cesarean delivery	78 (87.6)	84 (95.5)
Prenatal steroids (full course)	47 (52.8)	57 (64.8)
Premature rupture of membranes	39 (43.8)	39 (44.3)
Preeclampsia	9 (10.1)	8 (9.1)
Gestational diabetes	3 (3.4)	3 (3.4)
IGDM	1 (1.1)	2 (2.3)
*Smoking habits*		
Before pregnancy	19 (21.3)	14 (15.9)
During pregnancy	3 (3.4)	1 (1.1)
After pregnancy	1 (1.1)	2 (2.3)
Always	9 (10.1)	14 (15.9)
Age of mother at birth	32.5 [±5]	32.6 [±6.8]
*Neonatal parameters*		
Male sex	56 (62.9)	42 (47.7)
Gestational age (days)	190 [±16] − 27 + 1	190 [±16] − 27 + 1
Birth weight (g)	941 [±253]	932 [±256]
Small for gestational age	7 (7.9)	5 (5.7)
Gestational age (days) at discharge	265 [±12] − 37 + 6	265 [±15] − 37 + 6
Breast milk feeding at discharge	30 (33.7)	21 (23.9)
*Before discharge*		
Anemia	77 (86.5)	82 (93.2)
Number of PRBC	3.3 [±4]	3.1 [±4]
Erythropoietin therapy	62 (69.7)	75 (85.2)
*Neonatal morbidity*		
NEC grade I and II	4 (4.5)	0 (0)
PDA	34 (38.2)	33 (37.5)
ROP ≥ grade III	5 (5.6)	5 (5.7)
IVH grade I and II	9 (10.1)	4 (4.5)
IVH grade ≥ grade III	4 (4.5)	6 (6.8)
PVL	0 (0)	2 (2.3)

Categorical data are presented as numbers with percentages in round parentheses. Continuous data are presented as the mean ± standard deviation in squared parentheses. Anemia was defined as packed red blood cells or erythropoietin therapy needed. IVH—intraventricular hemorrhage, NEC—necrotizing enterocolitis, PDA—persisting ductus arteriosus, PRBC—packed red blood cells, PVL—periventricular leukomalacia, ROP—retinopathy of prematurity, SGA—small for gestational age (weight at birth <10th percentile).

**Table 2 nutrients-14-02732-t002:** Iron status.

Parameter	*6 Weeks Corrected Age*		*6 Months Corrected Age*		*12 Months Corrected Age*	
	Early Group (*n* = 89)	Late Group (*n* = 88)	Early Group (*n* = 89)	Late Group (*n* = 88)	Early Group (*n* = 89)	Late Group (*n* = 88)
Iron intake by supplements (mg/kg/d)	3.7 (3.3–4.0) *	3.1 (2.8–3.5) *	2.6 (2.3–2.9)	2.6 (2.3–2.8)	1.8 (1.2–2.5)	2.0 (1.5–2.5)
*Hematologic parameters*						
Ferritin (mcg/L)	47.1 (36.9–57.3)	46.5 (36.7–56.3)	30.5 (23.7–37.2)	34.2 (27.8–40.6)	31.7 (23.3–40.1)	30.0 (21.5–38.4)
Hemoglobin (g/dL)	10.9 (10.6–11.1) *	11.2 (11.0–11.4) *	12.2 (12.0–12.4)	12.3 (12.1–12.5)	12.3 (12.1–12.5)	12.3 (12.1–12.5)
Hematocrit (%)	31.4 (30.7–32.0)	32.3 (31.6–32.9)	34.6 (34.1–35.1)	34.9 (34.4–35.4)	35.3 (34.7–35.8)	35.3 (34.7–35.9)
RBC (/pl)	4.0 (4.0–4.1)	4.2 (4.1–4.3)	4.7 (4.6–4.8)	4.7 (4.6–4.8)	4.8 (4.7–4.9)	4.7 (4.6–4.8)
MCV (fl)	78.5 (77.5–79.4)	78.1 (77.3–79.0)	74.4 (73.5–75.3)	74.3 (73.5–75.2)	74.5 (73.5–75.4)	74.9 (73.9–75.9)
MCH (pg)	27.1 (26.8–27.5)	27.2 (26.9–27.5)	26.2 (25.8–26.6)	26.2 (25.9–26.6)	26.0 (25.5–26.4)	26.0 (25.6–26.5)
MCHC (g/dL)	34.6 (34.3–34.8)	34.8 (34.5–35.0)	35.2 (34.9–35.6)	35.1 (34.8–35.4)	34.8 (34.6–35.1)	34.8 (34.5–35.0)
Reticulocytes relative (‰)	17.0 (15.3–18.7)	15.3 (13.6–17.0)	10.8 (10.0–11.7)	10.5 (9.7–11.3)	10.0 (9.0–11.0)	9.4 (8.3–10.4)
Transferrin (g/L)	2.4 (2.3–2.5)	2.5 (2.4–2.6)	2.6 (2.5–2.7)	2.6 (2.5–2.7)	2.9 (2.8–3.0)	2.7 (2.6–2.8)
sTRF (mg/L)	1.6 (1.5–1.7)	1.7 (1.5–1.8)	1.7 (1.5–1.8)	1.6 (1.5–1.8)	1.7 (1.6–1.8)	1.8 (1.7–1.9)
Iron (mcg/dL)	85.4 (75.9–94.8)	89.1 (79.9–98.3)	70.5 (63.7–77.2)	72.9 (66.4–79.3)	60.7 (53.3–68.1)	64.4 (56.9–71.9)
Transferrin saturation (%)	25.4 (22.5–28.3)	26.2 (23.4–29.0)	19.1 (17.0–21.2)	20.2 (18.2–22.2)	15.4 (13.4–17.4)	16.9 (14.8–18.9)
*Iron deficiency and anemia*						
Iron deficiency	42/75 (56%)	48/76 (63%)	4/68 (6%)	5/67 (8%)	8/62 (13%) *	1/61 (2%) *
Anemia	1/88 (1%)	2/87 (2%)	3/79 (4%)	0/81 (0%)	0/74 (0%)	0/73 (0%)
Iron deficiency anemia	0/74 (0%)	2/76 (3%)	0/68 (0%)	0/67 (0%)	0/62 (0%)	0/61 (0%)

Data are presented as the estimated marginal mean and 95% CI in parentheses. Iron deficiency, anemia, and iron deficiency anemia are presented as number of patients and percentage in parentheses. *p* values < 0.05 were considered statistically significant, parameters with significant differences before correction for multiple testing were marked with *. After correction for multiple testing (Bonferroni), no significant differences were detected.

## Data Availability

The study protocol and the individual participant data that underlie the results reported in this article, after de-identification, are available upon request from the corresponding author 6 months after publication. Researchers will need to state the aims of any analyses and provide a methodologically sound proposal. Proposals should be directed to nadja.haiden@meduniwien.ac.at. Data requestors will need to sign a data access agreement and in keeping with patient consent for secondary use, obtain ethical approval for any new analyses.

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
