# Peer review of "Preterm Infants on Early Solid Foods and Iron Status in the First Year of Life—A Secondary Outcome Analysis of a Randomized Controlled Trial"

_nutrients, 2022, doi:10.3390/nu14132732_

Round 1

Reviewer 1 Report

This is an interesting study addressing the impact of timing of solid food introduction after discharge on iron status in infants born very preterm and with VLBW.

I only have few comments:

1. Please, explain the randomization procedure briefly. Did the randomization procedure take account of SGA status and feeding type (breast vs formula feeding)?

2. The number of SGA births is exceptionally low for a study using both a gestational age (<32 wks) and a birth weight (<1,500 g) based cut off. Could the authors explain this? Which reference curve was used to classify infants as SGA or AGA?

Reviewer 2 Report

This work shows no impact on iron status in the first year of life of very low birth preterm infants after the introduction of standardized solid foods in two timepoints (10th and 12th week and 16th and 18th week of life, early and late groups, respectively). This is a secondary outcome analysis of a prospective randomized clinical trial. Although the results are of interest for the readers, some concerns need to be addressed. 

Comment 1. Methods (lines 134-141): The wording used is not clear. Please rewrite these sentences.

Comment 2. There is no sample size or power calculation and the ratio of controls to case infants is 1:1. More details are needed. Please also provide arguments for the randomization and control/case ratio (1:1).

Comment 3. Did the authors perform the linear mixed effects model for the evaluation of group differences of the primary outcome ferritin taking into account the effect of red blood cell transfusion? This is not clear in the Methods section.

Comment 4. In Table S3, did the authors perform linear mixed models  for Bayley-III evaluation taking into account RBC transfusion, anemia, nutrition at discharge (breastfed vs. formula), cumulative iron supplementation adjusted for body weight, erythropoietin supplementation? Iron deficient infants had higher scores in Bayley-III test and this should be further discussed taking into account for other fixed parameters besides iron deficiency, sex, gestational age and multiple birth as stated in lines 153-155 of the Methods section. 

Comment 5. Please provide a conclusion in the abstract related to clinical importance of the results.
